# Imprinting of Mesenchymal Stromal Cell Transcriptome Persists even after Treatment in Patients with Multiple Myeloma

**DOI:** 10.3390/ijms21113854

**Published:** 2020-05-28

**Authors:** Léa Lemaitre, Laura Do Souto Ferreira, Marie-Véronique Joubert, Hervé Avet-Loiseau, Ludovic Martinet, Jill Corre, Bettina Couderc

**Affiliations:** 1UMR 1037 INSERM, 31059 Toulouse, France; lea.lemaitre@inserm.fr (L.L.); dosoutoferreira.laura@iuct-oncopole.fr (L.D.S.F.); marie-veronique.joubert@inserm.fr (M.-V.J.); avetloiseau.herve@iuct-oncopole.fr (H.A.-L.); ludovic.martinet@inserm.fr (L.M.); 2Université Paul Sabatier, Toulouse 3 University, 31062 Toulouse, France; 3Institut Universitaire du Cancer de Toulouse-Oncopole, 1 avenue Irene Joliot Curie, 31052 Toulouse, France; 4Institut Claudius Regaud, IUCT-O, 31052 Toulouse, France

**Keywords:** multiple myeloma, mesenchymal stromal cells, transcriptome, imprinting, adipogenesis, immunogenicity

## Abstract

Introduction. Multiple myeloma (MM) is a B-cell neoplasm characterized by clonal expansion of malignant plasma cells (MM cells) in the bone-marrow (BM) compartment. BM mesenchymal stromal cells (MSC) from newly diagnosed MM patients were shown to be involved in MM pathogenesis and chemoresistance. The patients displayed a distinct transcriptome and were functionally different from healthy donors’ (HD) MSC. Our aim was to determine whether MM–MSC also contributed to relapse. Methods. We obtained and characterized patients’ MSC samples at diagnosis, two years after intensive treatment, without relapse and at relapse. Results. Transcriptomic analysis revealed differences in gene expression between HD and MM-MSC, whatever the stage of the disease. An easier differentiation towards adipogenesis at the expense of osteoblatogeneis was observed, even in patients displaying a complete response to treatment. Although their transcriptome was similar, we found that MSC from relapsed patients had an increased immunosuppressive ability, compared to those from patients in remission. Conclusion. We demonstrated that imprinting of MSC transcriptome demonstrated at diagnosis of MM, persisted even after the apparent disappearance of MM cells induced by treatment, suggesting the maintenance of a local context favorable to relapse.

## 1. Introduction

Multiple myeloma (MM) is a hematological malignancy characterized by the abnormal expansion of clonal plasma cells [1]. Despite a huge improvement in survival in the last two decades, most MM patients relapse after a period of remission that is extremely variable from one patient to another. While this relapse is probably due to the selection of chemoresistant-MM cell clones [2,3], some authors have posited the influence of the microenvironment on the recurrence of MM [4,5].

MM develops in the bone marrow, which also contains mesenchymal stromal cells (MSC) that are non-hematopoietic multipotent progenitor cells. MSC are implicated in the pathogenesis of many solid cancers and hematological malignancies, including MM [6]. In MM, they are involved in the growth of malignant cells [7], the acquisition of chemoresistance [8,9], and in the abolition of the anti-tumor immune response [9]. They are involved in the different steps of tumor progression, in several ways [10]. They directly induce MM cell expansion through the huge amount of cytokines that they produce [6] and via the secreted exosomes [11]. They can dialogue directly with MM cells through close contact [7,12] and membrane exchange, inducing chemoresistance acquisition [13]. They also play a role in the recruitment of other microenvironment cells involved in tumor cell growth, chemoresistance acquisition, and depletion of the immune responses [14,15,16]

Our own studies and that of other authors have shown that MSC from patients with newly diagnosed MM are abnormal [11,17], as they acquire pro-tumor potentialities. For example, transcriptomic analysis showed that MSC from newly diagnosed MM patients expressed abnormally high levels of GDF15 and IL-6 [11], two cytokines implicated in MM cell growth and chemoresistance [18,19]. To determine whether MM BM-MSC also play a role in MM relapse, we analyzed the phenotype and transcriptome of MM BM-MSC throughout the different stages of the MM disease history—at diagnosis, during remission after intensive treatment, and at relapse. MSC are self-renewing and multipotent progenitors and as such present an asymmetric division with one cell retaining its strain character and one differentiating cell. Our aim was to determine whether the phenotypic modifications of MSC induced by the appearance of MM cells in the BM occur in the progenitor cells or the differentiated cells. In the first case, the acquisition of the MSC-associated cancer character would be perennial, whereas in the second, the cells would be expected to return to their original phenotype, when the patient is in remission with a complete response. Our second goal was to determine which MSC abnormalities could potentially be involved in MM relapse.

## 2. Results

### 2.1. Patients’ Samples

We performed this study by isolating MSC from newly diagnosed MM patients (D BM-MSC) (group 1, D BM-MSC, *n* = 12) or from patients treated for newly diagnosed MM, through first-line, high-dose melphalan with autologous stem cell transplantation (post treatment BM-MSC) (*n* = 19). Table 1 summarizes the patients’ characteristics. There were 8 men and 4 women in the diagnosis group, the median age at diagnosis was 51 +/− 6 years, and there were 6 men and 13 women in the post-treatment group and the median age at diagnosis was 59 +/− 5 years. MSC were isolated either at early relapse (ER BM-MSC) (group 2, *n* = 9) or in complete remission (CR BM-MSC) (group 3, *n* = 10). The bone marrow aspirations for the treated patients were performed 9 to 24 months post-transplant, with a median of 16 +/− 4 months for the ER BM-MSC group, 5 to 24 months for the post-transplant group, and 15 +/− 6 months for the CR BM-MSC group. Since January 1 2020, the patients in group 3 were still in complete remission, according to the consensus recommendations of the International Myeloma Working Group [20], between 4 and 6 years post-treatment, except for two patients who relapsed more than two years after the analysis (indicated with a star in Table 1). We used MSC from the healthy donors (HD BM-MSC) (group 4, *n* = 10) as the control cells.

Freshly collected MSC were selected by their plastic adherence and maintained in standard culture conditions, as described in the Material and Methods. Their identity was verified by flow cytometry analysis (Appendix A). HD and MM BM-MSC (including ER, CR, and D BM-MSC) displayed similar morphology with irregular shape and evidence of branching (data not shown). There were no significant differences in the percent expression of the MSC defining markers, between the HD and the MM BM-MSC, whatever the disease stage (Appendix A).

### 2.2. Transcriptomic Profile

To assess differences in gene expression between MSC isolated from the patients at early relapse (ER BM-MSC) or in complete remission (CR BM-MSC), an unbiased transcriptomic analysis was performed from the samples analyzed at passage 1 in culture. A heat map was generated, showing the raw-scaled log fragments per kilobase of exon model per million reads mapped (FPKM) +1 expression values. This process readily grouped the samples by type (ER vs. CR BM-MSC, Figure 1A). No gene was significantly differentially expressed between the two groups (*p* < 0.05). Whether the patient was in complete remission or at an early relapse stage had little impact on the MSC transcriptome.

Then, we compared the transcriptomic profile of MSC isolated from patients who had received their treatment (patients at ER or CR) to the transcriptomic profile of MSC from newly diagnosed patients or healthy donors. The objective was to determine whether MSC from the treated patients displayed a “normalized” transcriptome (i.e., similar or close to HD BM-MSC) or whether it was identical to the transcriptome of D BM-MSC. Figure 1B shows that two groups could be distinguished—HD BM-MSC and MM BM-MSC (including ER, CR and D BM-MSC), whatever the disease stage. D BM-MSC and MSC from the treated patients displayed a similar transcriptome and strongly differed from that of the HD BM-MSC. A total of 2747 genes were differentially expressed between HD BM-MSC and MM BM-MSC, at all stages, of which 440 had an adjusted *p*-value < 0.05 (lists of up- and downregulated genes can be provided upon request). Principal component analysis (PCA) confirmed the heatmap results 40.92% genes explained the differences between MM BM-MSC and HD BM-MSC and 7.06% explained the differences between D BM-MSC and the other three groups (Figure 1C). This suggests that the imprinting of the MSC transcriptome induced by MM cells persisted after intensive treatment, even for patients in complete remission.

We performed volcanoplots in order to represent genes that are differentially expressed in the four groups (Figure 2A, Figure 3A and Appendix A). Three groups could be distinguished based on the expression of 57% of the analyzed genes. The transcriptome of the treated patients’ MSC could be distinguished from that of D BM-MSC, since 1121 genes were differentially expressed between the two groups, with an adjusted *p*-value < 0.05 and 2 < Fold Change (FC) < 0.5. The transcriptome of the treated patients’ MSC was slightly different from that of D BM-MSC and was very different from that of HD BM-MSC. Therefore, there was no reversion of the MM BM-MSC phenotype after intensive treatment, even when the patients were in complete remission. In addition, intensive treatment induced an additional variation in the transcriptome of MM BM-MSC.

### 2.3. Signaling Pathways Activated in Patients with a History of Multiple Myeloma

Since MSC are known to be involved in the pathogenesis of MM [10,11] and because even patients achieving CR have a high probability of relapse, we focused on the common genes that were differentially expressed between the HD and MM patients, whatever the disease stage. One hundred seventy-four genes were differentially expressed between the HD BM-MSC and MM BM-MSC, and were common between the D BM-MSC, ER BM-MSC, and CR BM-MSC. Differentially expressed genes between HD BM-MSC and MM BM-MSC were analyzed with the Gene Set Enrichment Analysis (GSEA). This predicted dysregulation of four canonical pathways in the MM BM-MSC as compared to the HD BM-MSC. These included (1) the osteoblastogenesis pathway—reduced expression of *RUNX2*, *PDGFRL*, *WNT2B*, and overexpression of *DDK1* (Figure 2A); (2) the Wnt pathway—reduced expression of *WNT2B*, *WNT5A*, *WNT5B*, *SOX4*, *CD24*, *FZD3*, *RSPQ1*, *TCF7L2*, and *GPC4* and overexpression of *DKK1*, *ROR1*, *PCDH10*, *EDN1*, *CDH6*, and *CCND1* (Figure 2A,B); (3) the extracellular matrix organization (integrins)-encoded genes with a reduced expression of *ITGA11* and *COL12A1* and overexpression of *ITGA2* and *NTN4*; and (4) the coagulation pathway with an increased expression of *DPP4*, *IRGA2*, *CFH*, *F2RL*, *PLAT*, *SERPINB2*, and *TFPI* (Appendix A). We also evidenced dysregulation of several genes involved in the non-suppressive_T cell_versus_activated_T reg signaling pathway (Figure 3A), with a reduced expression of *ALX1*, *GFRA1*, *CXCL16*, *GPC4*, *GSTM4*, and *INHBE* and overexpression of *BDNF* and *DOK5* in the MM BM-MSC compared with HD BM-MSC (Figure 3A,B).

### 2.4. Functional Characterization of MM-MSC

MSC are known to promote the survival and even the growth of malignant plasma cells. We, therefore, compared the pro-survival activity of patient-derived MSC at different disease stages (D, ER, and CR BM-MSC) by performing co-culture experiments. No difference in pro-survival activity was observed between MSC groups.

Here, we have shown that the onset of the pathology induces a change in the MSC transcriptome that persists even after the curative treatment. However, MSC after treatment had a slightly different transcriptome than D BM-MSC. We aimed to determine whether this slight difference was towards normalization of the MSC phenotype or whether the treatment turned MSC into an even more pro-tumoral phenotype.

Since the osteoblast differentiation pathway and the Wnt pathway seemed to alternate (Figure 2A,B) and because many authors have reported the impaired osteoblastic potential of D BM-MSC [21,22,23,24], we compared the potential for adipocyte and osteoblast differentiation of MSC in the groups of patients, using RT-PCR analysis of specific gene expression, as described in the Materials and Methods section. Prior to this analysis, we used colorimetric tests to verify that post-treatment BM-MSC (ER and CR BM-MSC) were able to differentiate into adipocytes and osteoblasts [21]. Both post-treatment and HD BM-MSC differentiated into adipocytes (Figure 4A) and osteoblasts (Figure 4B,C), after 21 days of culture with the differentiation media. Post-treatment BM-MSC exhibited a significant increase in lipid accumulation between D0 and D21 of adipocyte differentiation (Figure 4A) and seemed to differentiate more likely into adipocytes than HD BM-MSC. On the other hand, the capacity for osteoblastic differentiation was impaired in post-treatment BM-MSC, including MSC from patients in complete remission (Figure 4B). Hence, post-treatment BM-MSC differentiated preferentially into adipocytes rather than osteoblasts, whatever the disease stage, thus, confirming the transcriptomic profile. To confirm this observation, we analyzed the expression of several genes involved in the multipotency of the MSC. Post-treatment BM-MSC overexpressed the genes implicated in adipocyte differentiation (PPARγ and adiponectin) (Figure 4D), while the expression of osterix, alkaline phosphatase, and Runx2 (involved in osteoblast differentiation) were down-regulated (Figure 4E). Of note, the increase in the expression of genes involved in adipogenesis was three-fold higher in post-treatment BM-MSC than in HD BM-MSC.

By analyzing the differentially activated signaling pathways between the transcriptome of the MSC groups, we also observed differences in the expression of the genes involved in the activation of the immune response (Figure 3A,B). *PTGS2* (COX2) expression was significantly upregulated in MM as compared to HD BM-MSC (FC = 2.03 and adj *p*-value = 0.12), in post-treatment BM-MSC (FC = 2.52 and adj *p*-value = 0.03), and was higher in ER BM-MSC compared to HD BM-MSC (FC = 2.95 and adj *p*-value = 0.03) (Figure 5A). We therefore compared the immunosuppressive potential of MM to HD BM-MSC. To address this question, MSC from the four groups were placed in contact with Peripheral Blood Mononuclear Cells (PBMC) at different ratios, and CD8 T-cell proliferation was analyzed (Figure 5B). The inhibitory effect of MSC on the T-cell proliferation occurred in a ratio-dependent manner. D BM-MSC tended to have a more immunosuppressive effect than those from HD, although the difference did not reach significance. The proliferation of CD8 T cells at 1/5 ratio (MSC to PBMC) was significantly inhibited by 57%, with ER BM-MSC, while proliferation at the same ratio was only inhibited by 23%, with CR BM-MSC. This immunosuppressive effect was also observed in ER BM-MSC but to a lesser extent than CR BM-MSC. The proliferation of T cells at 1/5 ratio (MSC to PBMC) was inhibited by 59% with D BM-MSC, while proliferation at the same ratio was inhibited only by 38% with HD BM-MSC (Figure 5C). Hence, the immunosuppressive effect was similar for D or ER BM-MSC, whereas it was similar for CR and HD BM-MSC.

## 3. Discussion

The issue of whether MSC are involved in the development of MM is now resolved. Several studies have reported the pro-tumoral role of MSC, whether through the growth factors or the exosomes they secrete, or by direct contact [6,7,8], but also through the recruitment of the immune system cells [25]. It is also known that D BM-MSC display an abnormal phenotype, when compared to HD BM-MSC [11,26]. However, whether the change in MSC phenotype involved the development of MM or whether it was solely a consequence of the disease remains to be determined. For this reason, several groups, including ours, have investigated the monoclonal gammopathy of undetermined significance (MGUS), bearing patients’ MSC. Indeed, all MM patients previously presented an MGUS, while 10 percent of people over 65-years old displayed an MGUS. We showed in a small cohort that some MGUS subjects displayed an MM BM-MSC phenotype and others displayed an HD BM-MSC phenotype [11]. Unfortunately, we were unable to conclude whether the abnormal phenotype was predictive of malignant progression.

Another important unsolved question was whether MSC played a role in the relapse process of MM. To address this issue, we compared CR BM-MSC to ER BM-MSC. We showed that the transcriptome of MSC from patients who had completed their treatment presented only very small differences from that of patients in apparent complete response and those relapsing. Two of the patients in the CR BM-MSC group relapsed a few months after the analysis, but no specificity in their transcriptome was detected. This suggests that the MSC phenotype cannot predict whether a patient will relapse.

MSC from patients who had completed treatment had a transcriptome that was essentially identical to that of D BM-MSC. Once modification of the MSC was achieved, the change was permanent, even when the patient was in remission after achieving a CR. This suggests persistent printing, whatever the disease stage. However, as significant progress was recently made for response evaluation in MM with the development of high-sensitivity minimal residual disease [27], it would be of interest to compare MSC from patients in CR with Minimal residual disease (MRD) < 10^−6^ and MSC from patients in CR with positive MRD. We wish we could have used a paired samples cohort (diagnosis-remission-relapse) from the same patients for this study, but this purpose was not realistic, given the current median progression free survival of the MM patients. We only studied one patient with two time points (diagnosis and remission); as shown in Appendix A, transcriptomes were almost identical, with no significant differences, according to the heatmap.

By analyzing genes that were differently expressed between the MSC of patients with a history of MM and those of HD, we highlight the activation of five signaling pathways in the MM microenvironment. These pathways were already mentioned by several authors as being involved in cancer [26,28,29,30,31].We were first interested in those already involved in MM pathogenesis, namely the osteoblastogenesis and Wnt signaling pathways [28]. Modulation of the Wnt/βcathenin signaling pathway in MSC was shown to induce a decrease in the expression of *RUNX2*, leading to osteoblastogenesis inhibition [24]. Liu et al. notably showed the involvement of adipocyte-secreted adipokines in MM progression and chemoresistance acquisition, via the upregulation of the expression of autophagic proteins in MM cells, leading to a suppression of the caspase cleavage and apoptosis in MM cells [9]. This was also observed in acute myeloid leukemia [32] and in solid tumors, such as prostate cancer [33]. We now showed that post-treatment BM-MSC did indeed differentiate more easily into adipocytes than into osteoblasts. We notably observed that expression of leptin was down-regulated in post-treatment BM-MSC, which was also in favor of the defect in osteogenesis [34]. The ability to better differentiate into adipocytes appeared as soon as MM was diagnosed. It persisted after treatment with variation in gene expression (Figure 4). The expression of adiponectin and PPARγ was significantly increased in MSC from the treated patients. This variation could be related to the effect of treatment [35]. Indeed, a change in the MSC phenotype, after chemotherapy, related to the entry of MSC into senescence was already reported [36].

On the one hand, the fact that MSCs differentiate more easily into adipocytes is in favor of a greater pro-tumoral effect [33]. Indeed, several studies have demonstrated the involvement of adipocytes in the progression of cancer, including MM [35,36,37]. The fact that MSC from patients are better differentiated into adipocytes is certainly in keeping with the pro-tumoral effect of MSC, although it is still too early to establish the link between this ability and relapse. The impairment of the osteoblastic defect that persists after treatment, even in patients in CR, could explain why osteoblastic lesions never heal [38].

It was also shown that the immunomodulatory effect of MSC might play a role in their pro-tumoral effect, by decreasing the anti-tumoral immune response mediated by CD8+ T-lymphocytes [39,40]. Kanamura et al. showed that MM is characterized by a defect in the recruitment and activation of CD8+ T-lymphocytes [41]. We therefore analyzed the immunosuppressive ability of MSC in the four groups and assessed the expression of genes involved in the LT activation. We found a difference in expression between HD and MM BM-MSC, with a significant MM overexpression of *BDNF* and underexpression of *ALX1*, *GFRA1*, *CXCL16*, *GSTM4*, *INHBE*, and *GPC4* (Figure 3B). Interestingly, the expression of *CXCL16* was different between early relapsed patients and those in CR BM-MSC. Concerning the immune checkpoint involved genes, we did not observe any significant difference between the different groups of samples. However, we did find a tendency of a higher expression of CD274 (PD-L1) in the MSC samples from CR BM-MSC, as compared to D and ER BM-MSC. These patients had a higher expression of CD86, CD48, and *HLADPB1*. The expression of CD40, *HLADRB1*, and PDL-2 (*PDCD1LG2*) showed no variation across the samples (Appendix A). Together with this transcriptome’s modification, we found a functional difference between the CR and ER BM-MSC, according to their immunosuppressive ability. CR BM-MSC behaved like HD BM-MSC, whereas ER BM-MSC behaved like D BM-MSC, i.e., with an increased immunosuppressive ability. However, we are unable at present to establish whether this functional transformation of MSC was a cause or a consequence of the growth of MM plasma cells. Overall, the presence of MM plasma cells in the BM induced changes, maybe epigenetic reprogramming, which switched the MSC towards a persistent pro-tumoral phenotype. These data suggest that MSC could play a role in the MM relapse, possibly by promoting the growth of the minimal residual cells.

## 4. Materials and Methods

### 4.1. Human Samples

#### 4.1.1. MSC Samples

Fresh BM aspirates from 31 patients with MM were collected at diagnosis (D BM-MSC), post-treatment, in complete response (CR BM-MSC) or at early relapse (ER BM-MSC), at the Institut Universitaire du Cancer de Toulouse-Oncopole through the Intergroupe Francophone du Myélome network. All patients gave written informed consent and collection was approved by the French Committee for the Protection of Persons (CPP; DC-2012-1654) on April 2012, as well as by the local IUCT-Oncopole review boards. Primary BM MM cells were purified by using magnetic anti-human CD138 microbeads (Miltenyi Biotec, Bergisch Gladbach, Germany). MM BM-MSC were obtained from the CD138-negative fraction and prepared, as previously described [11]. For the HD BM-MSC, the BM aspirations were harvested from healthy donors who gave their written informed consent, according to the recommendations of the Ethics Committee of the Toulouse University Hospital.

#### 4.1.2. PBMC Samples

Peripheral blood mononuclear cells (PBMC) obtained from the Establissement Français du Sang (Toulouse, France) were isolated by Ficoll-Hypaque 5GE (Healthcare) density centrifugation.

### 4.2. Isolation of BM MSC

Freshly collected MSC were selected by their plastic adherence and were maintained in standard culture conditions. After 21 days, MSC were analyzed through flow cytometry for surface antigen expression—monocyte markers (CD11b and CD14), plasma cell markers (CD138 and CD38), and MSC markers (CD73+ CD90+ CD105+ and CD45-). There were no significant differences in the percent expression of the MSC-defining markers between the HD and the MM BM-MSC, whatever, the disease stage (Appendix A). BM-MSC were separated into two groups, one of 2.10^5^ cells collected and frozen with 350 μL RLT (RNeasy Minikit Qiagen, Hilden, Germany) + 10% βmercaptoethanol (Sigma-aldrich, Missouri, USA) for transcriptomic analysis, and another which was expanded at 1000 cell/cm^2^ as P1 for functional analysis.

### 4.3. Transcriptomic Analysis

RNA from sorted MM and HD BM-MSC was extracted using the RNeasy Minikit (Ref: 74106, Qiagen, Hilden, Germany), according to the manufacturer’s instructions. Then, cDNA synthesis, in vitro transcription and fragmentation of cRNA were performed using the GeneChip 3′IVT PLUS Reagent kit, (Affymetrix, Santa Clara, CA, USA) according to the manufacturer’s instructions. After assessment of RNA integrity (Agilent Small RNA Analysis kits, Agilent 2100 Bioanalyser, Agilent, Santa Clara, CA, USA), biotinylated RNA was hybridized with the Affymetrix HG-U133 plus 2.0 GeneChip microarrays (Affymetrix, Santa Clara, CA, USA), and analysis was performed as described previously. Raw Affymetrix cell intensity files (.CEL) were used for the differential expression analysis.

### 4.4. Statistical Analysis

#### 4.4.1. PCA

Raw intensities were processed and normalized by functional robust multi-array average (RMA), using the Affy package from the R/Bioconductor. From the raw expression values, we performed a principal component analysis (PCA). The first dimension dissociated the samples left to right. HD BM-MSC clustered on the right and MM BM-MSC clustered on the left. The third dimension separated the diagnostics at the bottom and the ER and CR BM-MSC at the top.

#### 4.4.2. Heatmap

From these data, we retrieved the significant genes differentially expressed between the HD and the patients with a fold change between 2 and 0.5. We then plotted them as a heatmap to see the clustering between the four groups, using these differentially expressed genes, but also between the genes.

#### 4.4.3. GSEA

We performed Gene Set Enrichment Analysis (GSEA) to identify the enriched gene sets (FDR < 0.25, nominal *p*-value cutoff < 0.05). Enriched gene sets from (Gene Set 256: GO Wnt signaling pathway, Gene Set 28: Hallmark coagulation, and Gene Set 304: GSE15659 non-suppressive T-cell vs. activated T-reg) were plotted as a heatmap and enrichment plot, using the GSEA1.0 Broad institute software for R. Gene set hallmarks used for this analysis can be found in the GSEA MSig database (https://www.gsea-msigdb.org/gsea/msigdb/index.jsp). Here, we used the H, Hallmark gene sets, C2, the curated gene sets, C5, the GO gene sets, C6, the oncogenic gene sets, and C7, the immunological gene set.

#### 4.4.4. MSC Differentiation Assay

2.10^4^ cells/well were cultured in 24- or 6-well culture plates, with complete Minimum Essential Medium α (MEMα) (Thermofisher, Waltham, MA, USA) or with StemMACS AdipoDiff Media or StemMACS OsteoDiff Media (Miltenyi Biotec, Bergisch Gladbach, Germany), for adipocyte and osteoblast differentiation, respectively. The medium was changed twice a week. All 24-well cultures were stopped after 21 days, for colorimetric testing. Alizarin red staining—the cells were fixed with 70% ethanol, stained with 2% Alizarin Red for 10 min, washed with H_2_O, and analyzed. For quantification, the plates were thawed, distained by the addition of 800 μL of 10% acetic acid chloride monohydrate. The optical density was then measured at OD405 and the relative ratio of the cells cultured in the osteogenic conditions were determined, relative to the cells cultured in the stromal medium. [21]. For adipocyte differentiation, the Nile red (Sigma-aldrich, Missouri, USA) stain was assessed according to the manufacturer’s guidelines. Cultures in the 6-well plates were stopped after 14 days for the RT-PCR tests and the cells were harvested using trypsin with EDTA (Invitrogen, Carlsbad, CA, USA).

#### 4.4.5. Quantitative RT-PCR

The cells were lysed using the RNeasy Minikit (Qiagen, Hilden, Germany), followed by direct reverse transcription, using the SuperScript^TM^ VILO^TM^ Master Mix (Invitrogen, Carlsbad, CA, USA). Quantitative PCR (qPCR) was performed using the LightCycler^®^ 480 Probes Master on a LightCycler^®^ (Roche, Bâle, Swiss), using the Taqman probes (Thermofisher, Waltham, MA, USA) (Appendix A). The expression of individual genes was normalized to GAPDH, through the ΔΔCt method.

#### 4.4.6. Co-Culture of MSC and T-Lymphocytes

PBMC from HD were stained with the Cell Trace Violet (CTV) (Thermofisher, Waltham, MA, USA), prior to stimulation, followed by CTV dilution assessment, 5 days later. At day 0, they were mixed with MSC in a 50/50 MEMα and Rosewell’s Park Memorial Institute-1640 (RPMI) (Thermofisher, Waltham, MA, USA) supplemented with 10% FBS and 0.5% of ciprofloxacin complete medium. PBMC were seeded in triplicates at the concentration of 1.10^5^ cells/100μL RPMI/well, with various MSC concentrations (20 10^3^, 10 10^3^, 5 10^3^, or 0 MSC/100μL MEMα/well), and were stimulated with or without anti-CD2/CD3/CD28 microbeads (Miltenyi Biotec, Bergisch Gladbach, Germany). After 5 days in co-culture, all cells were harvested and the CD8 T-cells were analyzed by flow cytometry. CD8 T-cells were selected with a combination of different markers (viable cell, CD90-, CD73-, CD45+, CD3+, CD4-, CD8+) and CTV-negative was quantified to express cell proliferation. The percentage of inhibition was calculated as follows: 100 − (percentage of CD8 T-cell proliferation with MSC/percentage of CD8 T-cell proliferation without MSC) × 100.

## 5. Conclusions

The presence of MM plasma cells was already known to lead to a change in the phenotype of MSC. Here, we show for the first time that this change is perennial, since it lasts throughout the course of the disease, including complete remission and early relapse. Further analyses are required to identify new therapeutic targets from these pro-tumoral MSC.

## Figures and Tables

**Figure 1 ijms-21-03854-f001:**
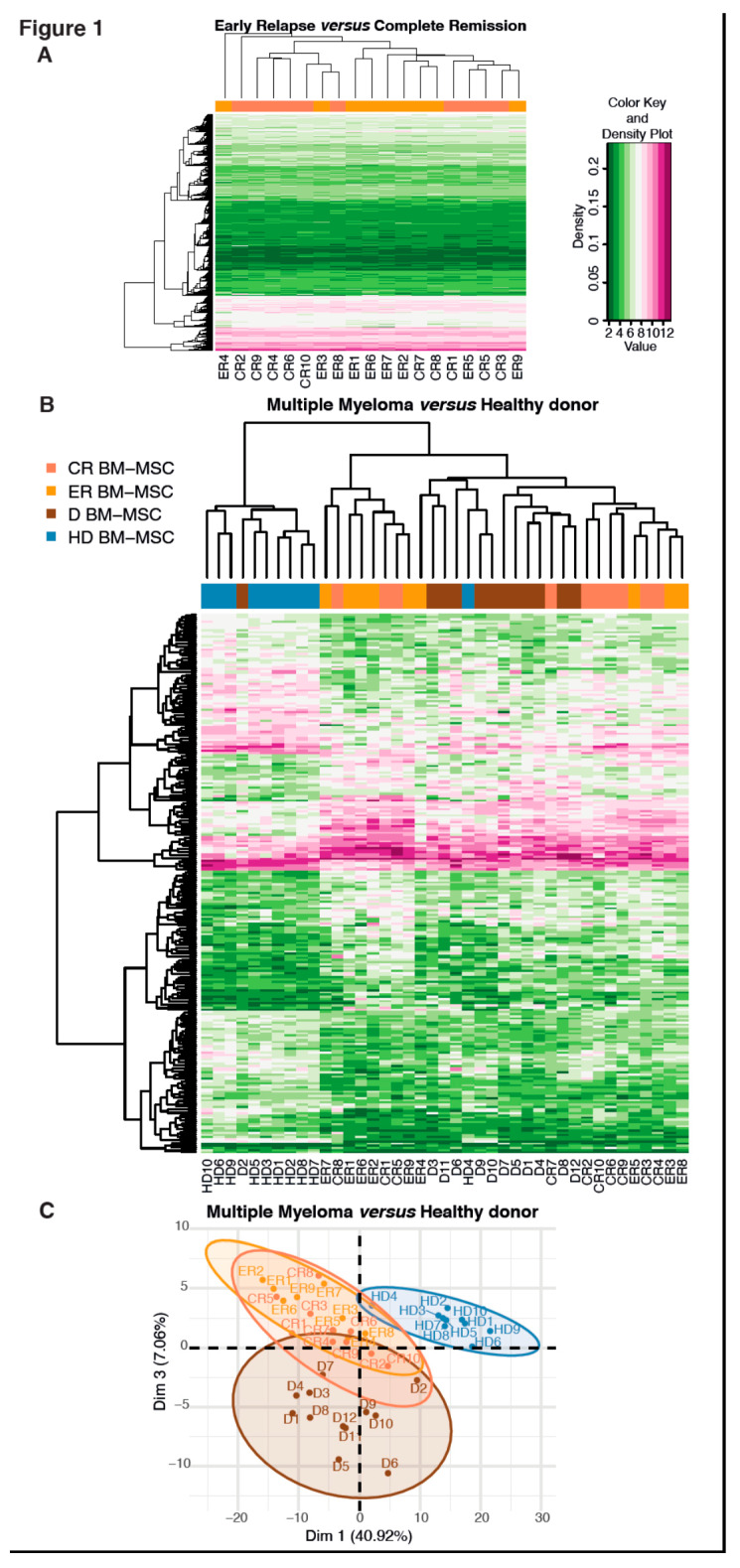
Transcriptional profile of ER and CR BM-MSC and clustering differences between HD and MM BM-MSC. (**A**) Unsupervised gene expression analysis of ER BM-MSC (*n* = 9) and CR BM-MSC (*n* = 10), representing a heatmap, with underexpression of green and overexpression of pink. No differences were observed between ER BM-MSC and CR BM-MSC, after the MM treatment. (**B**) Heatmap of 231 (108 upregulated in pink and 123 downregulated in green) differentially expressed genes in the MM BM-MSC (ER, CR and D BM-MSC) versus the HD BM-MSC. Expression data were filtered by the adjusted *p*-value < 0.05 and 2 < FC < 0.5. (**C**) Two-dimensional principal component analysis (PCA) of all MSC subtypes with representation of dimensions 1 and 3. There are three clusters based on the differences in their transcriptomic profiles. Expression data were filtered by the adjusted *p*-value < 0.05 and 2 < FC < 0.5.

**Figure 2 ijms-21-03854-f002:**
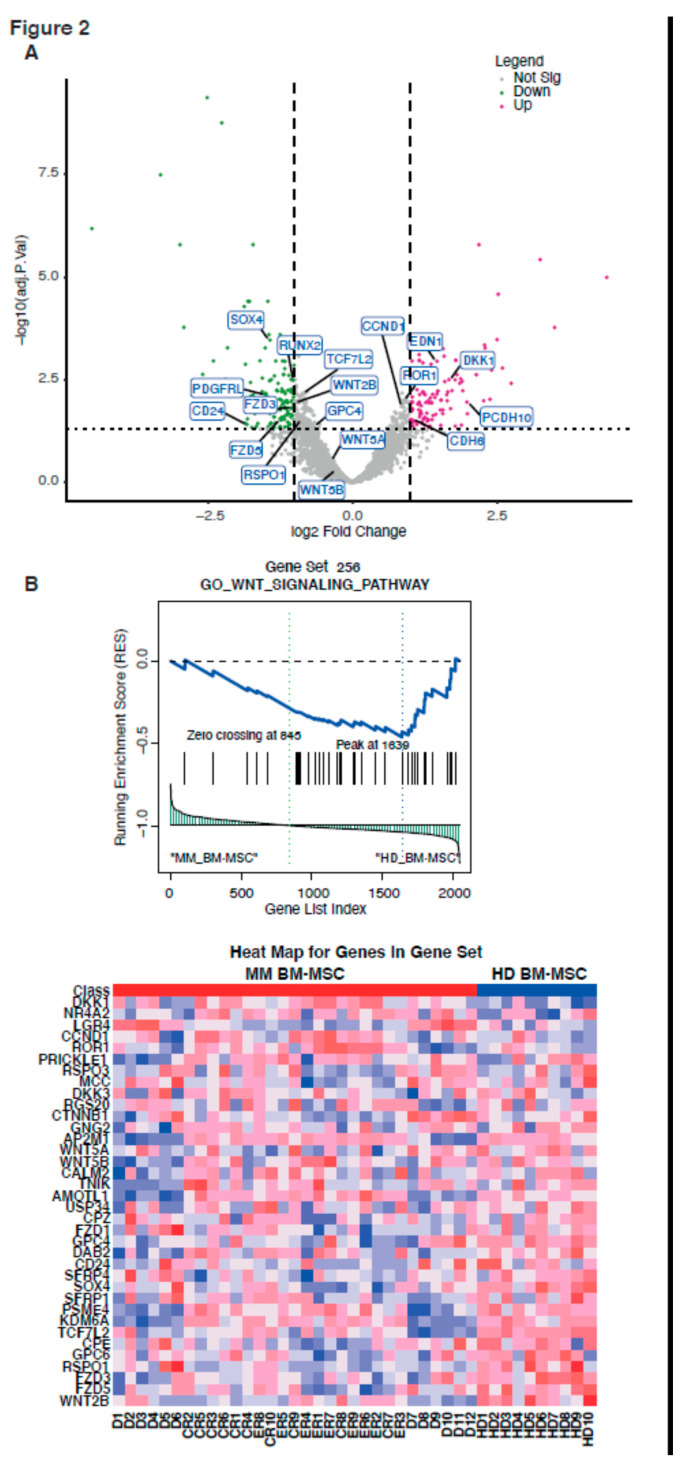
Wnt pathways involved in the difference between MM BM-MSC and HD BM-MSC. Volcano plot of differentially expressed genes in MM vs. HD BM-MSC with the impact of genes on osteogenesis and the Wnt pathway (**A**). Representative Gene Set Enrichment Analysis (GSEA) of the MM vs. HD BM-MSC list of expressed genes with the pathway of Gene Set 256 (GO Wnt signaling pathway). Wnt signaling pathway was downregulated by MM vs. HD BM-MSC. The corresponding heatmap of the represented genes is also shown (**B**).

**Figure 3 ijms-21-03854-f003:**
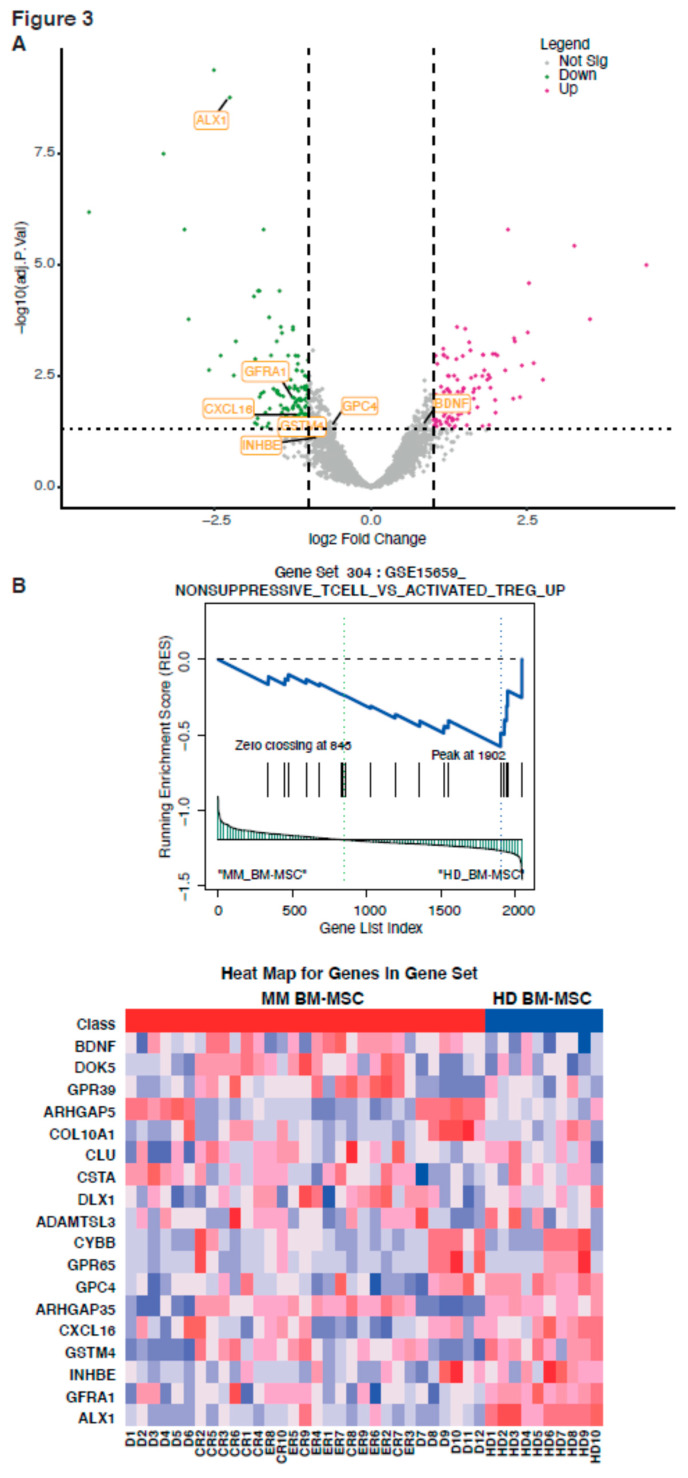
Immunity pathways involved in the difference between MM BM-MSC and HD BM-MSC. Volcano plot of the differentially expressed genes from MM vs. HD BM-MSC, with the impact of genes from the immunity pathway. (**A**) Representative GSEA of MM vs. the HD BM-MSC list of the expressed genes with the pathway of Gene Set 304 (GSE15659 non-suppressive T cell vs. activated T reg). Immunity pathway is downregulated by MM vs. HD BM-MSC. The corresponding heatmap of the represented genes is also shown (**B**).

**Figure 4 ijms-21-03854-f004:**
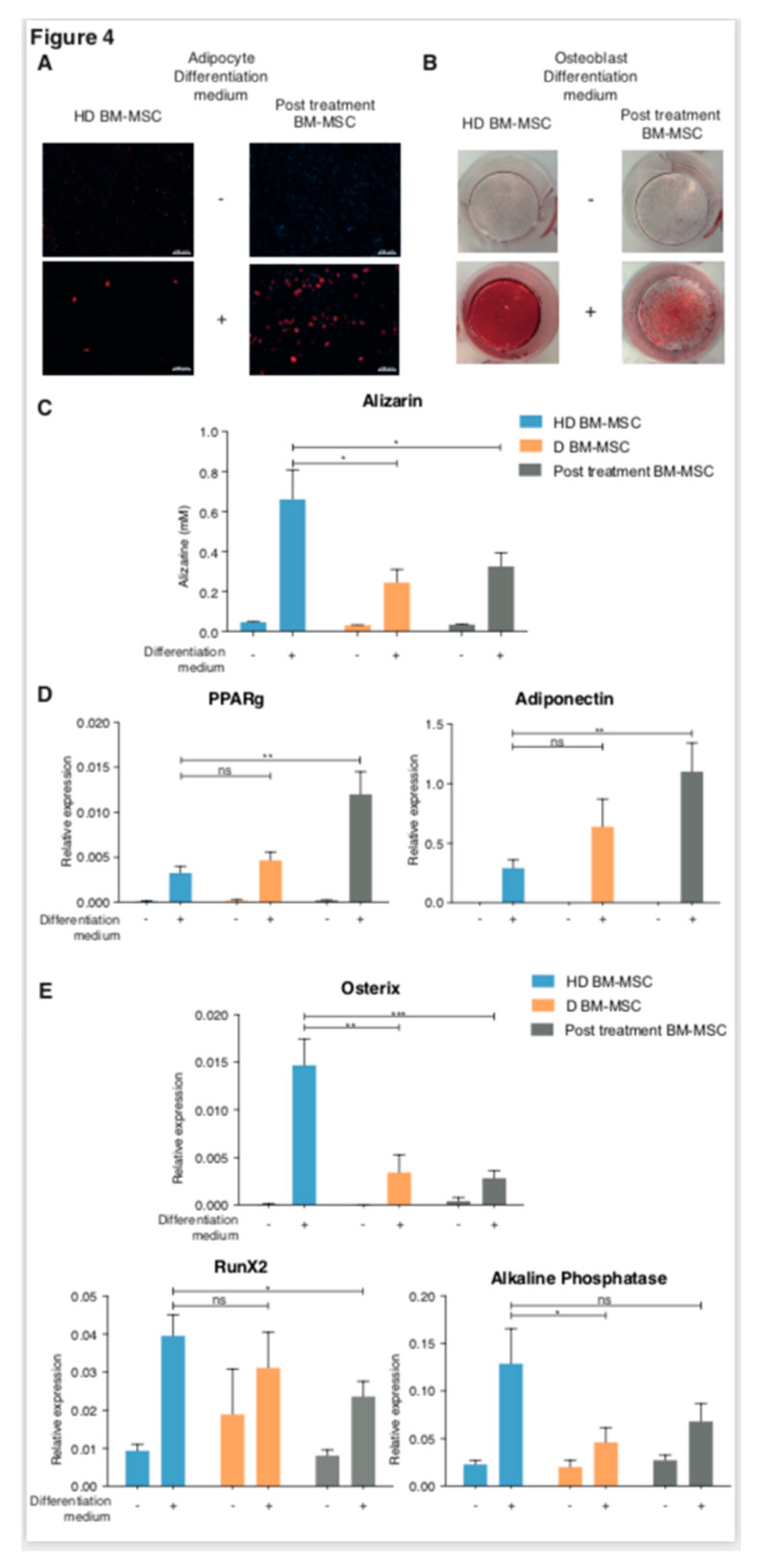
Colorimetric tests in adipocyte or osteoblast differentiation medium (**A**–**C**) or for 14 days before RT-qPCR analysis. (**D**,**E**) Colorimetric test was performed with Nile red for adipocyte (**A**) (magnification X5) and alizarin red for osteoblast (**B**,**C**). Alizarin Red Staining quantification using the assay. Data are expressed as the mean +/− SEM (HD BM-MSC *n* = 10; D BM-MSC *n* = 9, post treatment BM-MSC *n* = 17) with the error bars representing the standard deviations (**C**). RT-PCR of PPARgamma (PPARγ) and adiponectin were up regulated after adipocyte differentiation of post-treatment BM-MSC, compared to HD BM-MSC (**D**); and Osterix, Runx2, and Alkaline phosphatase were down regulated after osteoblast differentiation of post-treatment BM-MSC compared to HD BM-MSC (**E**). Data are mean ± SEM of the relative expression from seven independent experiments. * *p* < 0.05, ** *p* < 0.01, *** *p* < 0.001.

**Figure 5 ijms-21-03854-f005:**
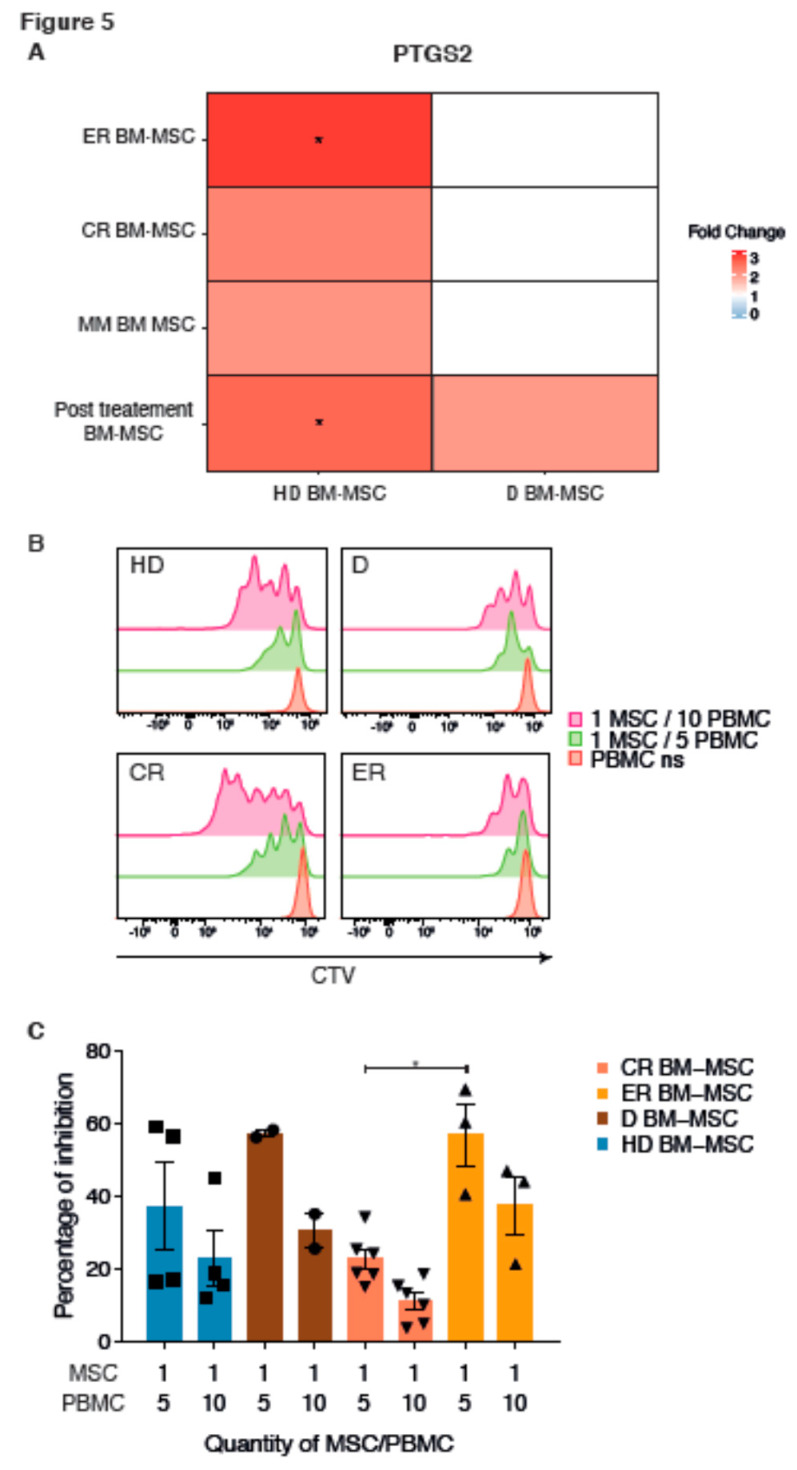
Modulation of immunity. Heatmap representation of *PTGS2* FC, between HD, D, ER, CR BM-MSC, or a combination of all MM (D, ER, and CR) BM-MSC, from red for upregulation to blue for downregulation. Groups on the left were compared to the bottom ones (**A**). Inhibitory effect of MSC from HD and MM BM-MSC on PBMC proliferation. Effect of MSC on cytotoxic CD8 T-cell proliferation. HD BM-MSC (*n* = 4), D BM-MSC (*n* = 2), CR BM-MSC (*n* = 6), and ER BM-MSC (*n* = 3) were cultured in the presence of increasing number of cell trace violet (CTV)-labeled PBMC for 5 days at the concentration of 1/5 or 1/10 MSC/PBMC, respectively. Then, all cells were harvested and analyzed by flow cytometry. Representative figure of different CD8 T-cell proliferation is shown with CTV. Non-stimulated PBMC in red represent the control of non-proliferation and was CTV-positive. 1MSC/5PBMC (green) and 1MSC/10PBMC (pink) show the CD8 proliferation inversely correlated with MSC quantity (**B**). Percentage of inhibition was calculated as follows: 100 − (percentage of CD8 T-cell proliferation with MSC/percentage of CD8 T-cell proliferation without MSC) ×100. Significant difference was observed between ER and CR BM-MSC. HD BM-MSC are representing in blue (■) D BM-MSC in brown (•), CR BM-MSC in salmon (▼) and ER BM-MSC in orange (▲). Data are expressed as mean ± SEM of the inhibition percentage of CD8 T lymphocytes. Each experiment was performed in triplicates from three independent experiments; (**C**). * *p* < 0.05.

**Table 1 ijms-21-03854-t001:** Patient characteristics.

	Patient Number	Male/Female	Age (Years)	Treatment	MSC Samples Obtained at (Months Post-Transplant)
**Early Relapse**	ER1	M	64	4VTD^1^/HD Mel + ASCT^2^/2VTD	14
	ER2	M	56	3VTD/HD Mel + ASCT/2VTD	9
	ER3	F	65	4VTD/HD Mel + ASCT/2VTD	23
	ER4	M	42	4VTD/HD Mel + ASCT/2VTD	18
	ER5	F	54	4VTD/HD Mel + ASCT/Unknown^3^	24
	ER6	F	50	4VTD/HD Mel + ASCT/2VTD	12
	ER7	F	61	4VTD/HD Mel + ASCT/2VTD	12
	ER8	F	59	xVTD/HD MEL + ASCT/Unknown	15
	ER9	F	64	4VTD/HD Mel + ASCT/xVTD	19
**Complete Remission at time of analysis**	CR1*	M	54	xVTD/HD Mel + ASCT/Unknown^3^	13
	CR2	F	65	4VTD/Unknown	22
	CR3	F	62	xVTD/HD Mel + ASCT/Unknown	5
	CR4	F	52	4VTD/HD Mel + ASCT/xVTD	6
	CR5*	F	65	4VTD/HD Mel + ASCT/Unknown	24
	CR6	M	67	xVTD/HD Mel + ASCT/Unknown	14
	CR7	F	64	4VTD/HD Mel + ASCT/xVTD	12
	CR8	F	63	4VTD/HD Mel + ASCT/Unknown	21
	CR9	F	59	5VTD/HD Mel + ASCT/2VTD	8
	CR10	M	56	4VTD/HD Mel + ASCT/Unknown	24

^1^ Velcade Thalidomide Dexamethasone, ^2^ High-Dose Melphalan + Autologous Stem Cell Transplantation, ^3^ Unknown Number of cycles, * Patients relapsing 2 years post sampling. Healthy donors (6 males and 4 females) were from 20 to 34 years old. Patients at diagnosis (8 males and 4 females) were from 37 to 59 years old.

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
