# Peer review of "Imprinting of Mesenchymal Stromal Cell Transcriptome Persists even after Treatment in Patients with Multiple Myeloma"

_ijms, 2020, doi:10.3390/ijms21113854_

Round 1

Reviewer 1 Report

In their manuscript entitled “Imprinting of mesenchymal stromal cell transcriptome persists even after treatment in patients with multiple myeloma“ Lemaitre at al. analyze the transcriptome of MSC in MM patients at different stages of disease as well as functional correlation.

Their results show the strong similarities among the MSCs despite different disease stages, thus suggesting a persistent imprinting independent of disease burden. Interestingly, the only difference noticed is in the immune regulation ability of these cells at time of relapse compared to complete remission.

The paper is well-written, the experiments are clearly described, results are consistent with previous publications and provide new interesting information.

Major

- The main flaw of the project is the cross-sectional rather not prospective design. Follow-up studies should focus on transcriptome changes in MSCs from the same patient before and after treatment.

- A better description of patients´ characteristics at relapse would help understand the differences in the immunosuppressive ability between CR-MSC and ER-MSC. In particular, which was the percentage of MM cell infiltrate in the samples at relapse (ER-MSC)? Do the % of MM cells correlate with immunosuppression results (the more tumor cells the more immunosuppression or viceversa)?)

- In Figure 4A and 4B D BM-MSC should be added. In addition, I recommend to quantify the inhibition of alizarin red.

- Figure 4D: osterix and runx2 mRNA levels should be assessed at an earlier time point (day 3 to day 7 of differentiation).

- Did the authors check at the expression of PD-L1 or other immunomodulatory molecules on MSCs?

Minor

Grammatical and typing errors such as

- Table 1 patient CR2 4VT/greffe/NR

- Page 12 line 12 “significantly buy post treatment”

- Page 13 line 3 “combinaison”

- Paragraph 5.5: please rewrite the OB differentiation part! (line 29: refsteoDiff or line 32 assessed by alizarine red staining alkaline phosphatase activity….)

References should be carefully revised: ref 21 does not refer to the IMF guidelines (page 2 line 39) or ref. 22 and 23 are wrongly cited on page 14 line 10.

Author Response

Answers to reviewers

First of all we would like to thank the reviewers for the time spent on our manuscript and for the remarks and questions asked which will allow us to improve our manuscript.

You will find a version of the manuscript where the corrections are apparent so that reviewers can appreciate the changes we have made to the manuscript. (LemaitreIJMSrevisedwithmarks) as well as the version where the changes are accepted.

We hope we have met the reviewers' expectations and remain at their disposal for further information if needed.

Thank you in advance for your attention

With kind regards

Bettina Couderc

Answers to the reviewers' questions point by point

Reviewer 1

Major revisions : The main flaw of the project is the cross-sectional rather not prospective design. Follow-up studies should focus on transcriptome changes in MSCs from the same patient before and after treatment.

We wish we could have used a paired samples cohort (diagnosis-remission-relapse) from the same patients for this study, but this purpose was not realistic given the current median progression free survival of MM patients. We only studied one patient with two disease course time points (diagnosis and remission); as shown in Figure S3, transcriptomes were almost identical, with no significant differences according to heatmap : this observation comforts the data we obtained in non paired samples. We have now added this issue in Discussion section. We have added in the new version of the manuscript the transcriptomes of the sample matched pair (diagnosis and remission). We have mentioned in the discussion this analysis, the results of which support our conclusions (last paragraph page 13 of LemaitreetalIJMSrevisedwithmarks).

- A better description of patients´ characteristics at relapse would help understand the differences in the immunosuppressive ability between CR-MSC and ER-MSC. In particular, which was the percentage of MM cell infiltrate in the samples at relapse (ER-MSC)? Do the % of MM cells correlate with immunosuppression results (the more tumor cells the more immunosuppression or viceversa)?)

We totally understand the reviewer question and we agree that it could be theorically very interesting to search for such a correlation. Unfortunately in practise, french clinicians don’t perfom bone marrow biopsies but only bone marrow aspirations, which are often diluted with blood (at all degree) ; consequently, bone marrow plasma cell percentages does not correctly reflect tumor burden. Hence, most of studies with bone marrow aspirations failed to demonstrate a prognostic value of bone marrow plasma cells percentage. Some myeloma with 10% are very agressive and others with 30% can be smoldering. However, we agree with the reviewer that immunosuppression degree may reflect stage of the disease (our data strongly support this hypothesis), but we believe the mechanism is multifactorial, and may be also modulated by genomic plasma cells abnormalities. In this study, our aim was above all to describe how MSC evolve during disease course.

- In Figure 4A and 4B D BM-MSC should be added. In addition, I recommend to quantify the inhibition of alizarin red.

The reviewer will notice in the new version that we have added D BM-MSC in Figure 4. We have performed the quantification of alizarin red inhibiton. This quantification is added  Figure 4C. The material and methods section has been modified to reflect the change in the figure (paragraph MSC differentiation page 17 of the LemaitreIJMSrevisedwithmarks version ) and the figure legend.

- Figure 4D: osterix and runx2 mRNA levels should be assessed at an earlier time point (day 3 to day 7 of differentiation).

We apologize but we cannot provide these results. Indeed we can not within the time frame for the review put cells back into culture and redo the experiment on cells. We apologize for this.

Did the authors check at the expression of PD-L1 or other immunomodulatory molecules on MSCs?

We analyzed the expression of genes involved in immunostimulation in our different MSC groups. You will find attached a vulcanoplot (supplemental figure 4) where they are represented some genes. We do not observe any significant difference between the different groups of samples. However, we did find a higher expression of CD274 (PD-L1) in MSC samples from patients in remission compared to patients at diagnosis or in relapse. These patients had higher expression of CD86, CD48 and HLADPB1. The expression of CD40, HLADRB1 and PDL-2 (PDCD1LG2) showed no variation across the samples. These observations have been included in the new version of the manuscrit first paragrah page 15 of the revised manuscript with marks.

Minor

Grammatical and typing errors such as

- Table 1 patient CR2 4VT/greffe/NR

- Page 12 line 12 “significantly buy post treatment”

- Page 13 line 3 “combinaison”

We corrected typos and grammar errors. You will see in the version of the manuscript with marks the corrections made.

- Paragraph 5.5: please rewrite the OB differentiation part! (line 29: refsteoDiff or line 32 assessed by alizarine red staining alkaline phosphatase activity….)

The paragraph has been reworded. It now reads:

2.104 cells/well were cultured in 24- or 6-well culture plates with complete MEMα or with StemMACS AdipoDiff Media or StemMACS OsteoDiff Media (Ref: 130-091-677 and 130-091-678, MACS Miltenyi) for adipocyte and osteoblast differentiation, respectively. Medium was changed twice a week. 24-well cultures were stopped after 21 days for colorimetric testing. Alizarin red staining: cells were fixed with 70% ethanol, stained with 2% Alizarin Red for 10 min, washed with H2O and analyzed. For quantification plates were thawed, destained by the addition of 400 μL of 10% cetylpyridinium chloride monohydrate. The optical density of the eluates was then measured at OD540 with a Bio-Rad plate reader and the relative ratio of the cells cultured in osteogenic conditions determined relative to cells cultured in stromal medium. [21]. For adipocyte differentiation, Nile red (Sigma, France) stain was assessed according to the manufacturer’s guidelines. Cultures in 6-well plates were stopped after 14 days for RT-PCR tests and cells were harvested using trypsin with EDTA (Invitrogen, ThermoFisher).

References should be carefully revised: ref 21 does not refer to the IMF guidelines (page 2 line 39) or ref. 22 and 23 are wrongly cited on page 14 line 10.

We apologize for the error in the bibliographical references. In fact two zotero libraries had been merged into the previous version. We have corrected and the following missing reference has been added.

Consensus recommendations for the uniform reporting of clinical trials: report of the International Myeloma Workshop Consensus Panel 1Clinical Trials & Observations

  1. Vincent Rajkumar , Jean-Luc Harousseau , Brian Durie , Kenneth C. Anderson , Meletios Dimopoulos , Robert Kyle , Joan Blade , Paul Richardson , Robert Orlowski , David Siegel , Sundar Jagannath , Thierry Facon , Hervé Avet-Loiseau , Sagar Lonial , Antonio Palumbo , Jeffrey Zonder , Heinz Ludwig , David Vesole , Orhan Sezer , Nikhil C. Munshi , Jesus San Miguel , on behalf of the International Myeloma Workshop Consensus Panel 1 Blood (2011) 117 (18): 4691–4695. https://doi.org/10.1182/blood-2010-10-299487

Reviewer 2 Report

Lemaitre et coauthors define the transcriptional landscape of mesenchymal stromal cells of multiple myeloma patients at different stages of the disease course (newly diagnosed and after therapy) and with different outcome after therapy (complete response, early relapse).  As a control, they analyzed healthy donor MSC. The major findings of the research are the clear difference of the transcriptome between MM and HD MSC. Differences between ND-MM and after treatment. Not much changes according to the type of evolution after therapy. Changes in signaling involved in the immunosuppressive milieu. Post treatment MSC impairment in osteoblast differentiation.

The paper deserves merit since its findings may have implications for the therapy of MM.

The major limitations are the heterogeneous nature of the sample studied and the lack of a intra-patient tracking of the MSC changes over time. If this could be done, even if for few cases, it would add much to the robustness of the whole study. 

Author Response

Answers to reviewers

First of all we would like to thank the reviewers for the time spent on our manuscript and for the remarks and questions asked which will allow us to improve our manuscript.

You will find a version of the manuscript where the corrections are apparent so that reviewers can appreciate the changes we have made to the manuscript. (LemaitreIJMSrevisedwithmarks) as well as the version where the changes are accepted.

We hope we have met the reviewers' expectations and remain at their disposal for further information if needed.

Thank you in advance for your attention

With kind regards

Bettina Couderc

Answers to the reviewers' questions point by point

Reviewer 2

The major limitations are the heterogeneous nature of the sample studied and the lack of a intra-patient tracking of the MSC changes over time. If this could be done, even if for few cases, it would add much to the robustness of the whole study. 

We wish we could have used a paired samples cohort (diagnosis-remission-relapse) from the same patients for this study, but this purpose was not realistic given the current median progression free survival of MM patients. We only studied one patient with two disease course time points (diagnosis and remission); as shown in Figure S3, transcriptomes were almost identical, with no significant differences according to heatmap : this observation comforts the data we obtained in non paired samples. We have now added this issue in Discussion section. We have added in the new version of the manuscript the transcriptomes of the sample matched pair (diagnosis and remission). We have mentioned in the discussion this analysis, the results of which support our conclusions (last paragraph page 13 of LemaitreetalIJMSrevisedwithmarks).

Round 2

Reviewer 1 Report

Revisions are fine, data are relevant and worth publication.